# Tensorpac: An open-source Python toolbox for tensor-based phase-amplitude coupling measurement in electrophysiological brain signals

Etienne Combrisson[1,2]*, Timothy Nest[1,3], Andrea Brovelli[2], Robin A. A. Ince[4], Juan L. P. Soto[5], Aymeric Guillot[6], Karim Jerbi[1,7,8]*

**1** Psychology Department, University of Montréal, QC, Canada, **2** Institut de Neurosciences de la Timone, UMR 7289, Aix Marseille Université, CNRS, 13385 Marseille, France, **3** Département d'informatique et de recherche opérationnelle, University of Montréal, QC, Canada, **4** Institute of Neuroscience and Psychology, University of Glasgow, Glasgow, UK, **5** Telecommunications and Control Engineering Department, University of Sao Paulo, Sao Paulo, Brazil, **6** Univ. Lyon, UCBL-Lyon 1, Laboratoire Interuniversitaire de Biologie de la Motricité, EA 7424, F-69622 Villeurbanne, France, **7** MEG Center, University of Montréal, QC, Canada, **8** Mila - Quebec Artificial Intelligence Institute, QC, Canada

\* e.combrisson@gmail.com (EC); karim.jerbi@umontreal.ca (KJ)

**Data Availability Statement:** The data used to showcase and validate the implemented functionalities of the software are generated on the

## Abstract

Despite being the focus of a thriving field of research, the biological mechanisms that underlie information integration in the brain are not yet fully understood. A theory that has gained a lot of traction in recent years suggests that multi-scale integration is regulated by a hierarchy of mutually interacting neural oscillations. In particular, there is accumulating evidence that phase-amplitude coupling (PAC), a specific form of cross-frequency interaction, plays a key role in numerous cognitive processes. Current research in the field is not only hampered by the absence of a gold standard for PAC analysis, but also by the computational costs of running exhaustive computations on large and high-dimensional electrophysiological brain signals. In addition, various signal properties and analyses parameters can lead to spurious PAC. Here, we present Tensorpac, an open-source Python toolbox dedicated to PAC analysis of neurophysiological data. The advantages of Tensorpac include (1) higher computational efficiency thanks to software design that combines tensor computations and parallel computing, (2) the implementation of all most widely used PAC methods in one package, (3) the statistical analysis of PAC measures, and (4) extended PAC visualization capabilities. Tensorpac is distributed under a BSD-3-Clause license and can be launched on any operating system (Linux, OSX and Windows). It can be installed directly via pip or downloaded from Github (https://github.com/EtienneCmb/tensorpac). By making Tensorpac available, we aim to enhance the reproducibility and quality of PAC research, and provide open tools that will accelerate future method development in neuroscience.

fly (not loaded from a repository). The exact same data can be accessed through the scripts that replicate the paper figures. These have been made available here: https://github.com/EtienneCmb/tensorpac/tree/master/paper.

**Funding:** EC and AB were supported by the French National Agency (ANR-18-CE28-0016-01) (http://anr.fr). EC was also supported by funding via a Natural Sciences and Engineering Research Council of Canada (NSERC) (https://www.nserc-crsng.gc.ca). JLPS acknowledge support from the Brazilian Ministry of Education (CAPES grant 1719-04-1) and the Fulbright Commission to JLP Soto (https://www.iie.org/programs/capes). KJ was supported by funding from the Canada Research Chairs program and a Discovery Grant (RGPIN-2015-04854) from NSERC (Canada), a New Investigators Award from FQNT (2018-NC-206005) and an IVADO-Apogée fundamental research project grant (http://www.frqnt.gouv.qc.ca/). This research is also supported in part by the FRQNT Strategic Clusters Program (2020-RS4-265502 - Centre UNIQUE - Union Neurosciences & Artificial Intelligence - Quebec) (https://ivado.ca/). TC acknowledges support through the Centre de Recherches Mathématiques (CRM). RAAI was supported by the Wellcome Trust (214120/Z/18/Z) (https://wellcome.ac.uk/). The funders had no role in study design, data collection and analysis, decision to publish, or preparation of the manuscript.

**Competing interests:** The authors have declared that no competing interests exist.

This is a *PLOS Computational Biology* Software paper.

## Introduction

The study of electrophysiology is innately challenging due to the immense complexity of oscillatory phenomena organized at many distinct spatial and temporal scales. While common assays for measuring brain function like fMRI are able to considerably reduce the temporal complexity of functional brain dynamics, scientists interested in electrophysiology must grapple with a dizzying array of plausibly meaningful features in the spectral domain. For decades, neuroscientists have sought to isolate cognitive and task-related changes in brain oscillations by examining spectral features such as power, amplitude, and phase across frequencies and brain regions. However, increasing attention has been given to more complex and dynamic properties of neural oscillations [1–6]. A prominent example of such dynamic oscillatory phenomena is Cross-Frequency Coupling (CFC) [7] which has been observed both at the phase-level [8–10], and at the amplitude level [11–13]. A slightly more recent, and arguably less well characterized phenomenon, Phase-Amplitude Coupling (PAC), provides a metric to identify and quantify synchronization between the phase of low-frequency oscillations and the amplitude of high-frequency oscillations.

Over the last decade, PAC has been shown to mediate a variety of task-related and cognitive functions including attention and decision-making [14, 15], learning and memory [16–22], motor and visuomotor tasks [10, 23–29], as well as mental disorders such as Parkinson disease and schizophrenia [30–37]. It has been proposed that PAC reflects the regulation of high frequency local computations by a larger network, oscillating at lower frequencies [38]. PAC might therefore contribute to coordinate neural activity by using a "hold and release" mechanism of gamma oscillations [29].

In order to quantify PAC, a number of methodologies and implementations have been proposed [14, 35, 39–46] and compared [45, 46]. Until now, there is still no gold standard on which method is the best alternative, even though the Modulation Index [46] is probably the most widely adopted due to its noise tolerance and amplitude independence. In addition, it has been shown that PAC can be computed in an event-related manner [47]. Nevertheless, there is still no consensus on the minimal data length (i.e. the number of cycles) that is required or the most appropriate filtering methods [48, 49]. It has also recently been shown that spurious PAC can occur for a variety of reasons that may be difficult to systematically control [48, 50–52]. Among them we can mention the absence of a clear peak in the power spectrum density (PSD) of the phase, the choice of the filter bandwidth or the nonstationarity.

To date, a handful of established brain data analysis toolboxes provide built-in functionalities to compute PAC, these include Matlab-based packages such as Fieldtrip [53], Brainstorm [54] and EEGLAB [55]. The open-science Python research community has also proposed a few PAC tools, including pacpy (https://github.com/voytekresearch/pacpy/) or pactools [41], with the latter supporting MNE-Python inputs [56]. While the available tools are extremely valuable, and although some of them provide multiple methods for PAC computation, existing tools do not always include time-resolved PAC measurements and often have limited options for visualization and statistical assessment. In addition, computational time remains a challenge that can severely limit PAC analyses at a time when data dimensionality and data-driven analyses are drastically increasing in neuroscience research.

Here, we present Tensorpac, a cross-platform open-source Python toolbox, distributed under a BSD-3-Clause license, dedicated to the measurement of phase-amplitude

relationships. This includes an array of functions to compute PAC and event-related PAC (ERPAC) alongside innovative features such as the estimation of the preferred-phase with polar plotting and exhaustive exploratory analysis across the full frequency space. Tensorpac also ships with additional tools in order to assess the reliability of the estimation such as power spectral density (PSD), Inter-Trial Coherence (ITC) and statistics. Crucially, what distinguishes Tensorpac even more from other available tools, is the combination of parallel computing and tensor-based implementation of the algorithms which drastically reduces computation time and opens up the possibility to compute PAC on large multidimensional arrays.

## Design, implementation and results

In principle, estimating PAC consists in quantifying the coupling between slow-wave phase with the amplitude of higher frequency signals. As a bidirectional coupling measure, however, it is impossible to say whether PAC high-amplitude rhythms are led by slow oscillations or the contrary. Accordingly, we denote by $f_1 \leftrightarrow f_2$ the PAC between a phase centered in $f_1$ and the amplitude centered in $f_2$.

### Estimation of a corrected phase-amplitude coupling

Estimating PAC is usually assessed in four steps, illustrated in Fig 1. First, the instantaneous phases of slower oscillations and amplitudes of faster oscillations are extracted. Second, the true coupling measure between those phases and amplitudes is computed. Third, a null distribution of surrogate values of the measure in the absence of coupling is estimated. This is usually assessed by swapping either amplitude or phase time-blocks, cut at a random time-point. Finally, the true coupling measure is corrected by subtracting the mean and dividing by the standard deviation of the surrogate null distribution. This step improves the robustness and the sensibility of the measure.

**Generating coupled signals.**    For the implementation and validation of coupling methods, we used synthetic signals with controllable coupling frequencies. To this end, we included in the toolbox two ways to simulate synthetic signals that can be imported from *tensorpac.signals*: *pac_signals_tort* which is a method based on pure sines summation and modulation [46] and *pac_signals_wavelet* which extract the phase from a random distribution leading to more complex signals [41]. Both methods, illustrated in Fig 2, provide fine-grained control over the coupling frequency pair of (phase, amplitude), the amount of coupling and noise such as data length and sampling frequency.

**Extracting the instantaneous phase and the amplitude.**    Since there is no consensus about whether the Hilbert or wavelet transforms constitutes a gold standard for extracting the phase and the amplitude, Tensorpac implements both. The Hilbert transform has to be applied on pre-filtered signals. For filtering, we implemented a Python equivalent to the two-way zero-phase lag finite impulse response (FIR) Least-Squares filter implemented in the EEGLAB toolbox [55]. Filter orders are frequency dependent and are defined as a function of the number of cycles (by default 3 cycles are used for the phase and 6 cycles for the amplitude [57]). The phase and the amplitude are respectively obtained by taking the angle and the absolute value of the Hilbert transform applied to the complex analytic filtered signals. Both components can also be obtained by convolving with a Morlet's wavelets [58] with a default width of 7, a default value broadly used in electrophysiological data analyses.

**Implemented PAC methodologies.**    Here, we describe the main PAC estimation methods currently available in Tensorpac, which include the Mean Vector Length, Modulation Index, Height-Ratio, normalized direct-PAC, phase-locking value and Gaussian-copula PAC (new

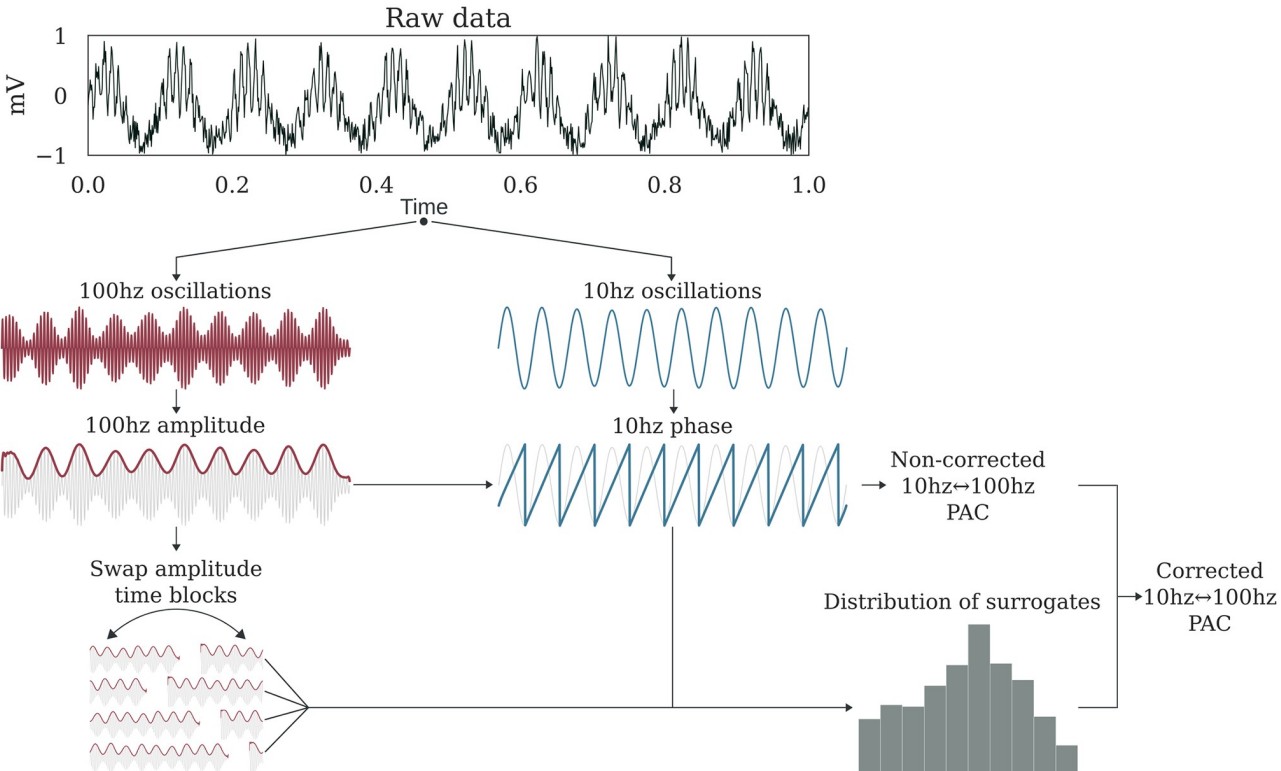

**Fig 1. Estimation process of a corrected 10↔100hz PAC.** For illustration, here simulated raw data contains a coupling between a 10 hz phase and a 100 hz amplitude. First, the raw data is respectively filtered with bandpass filters centered on 100hz and 10hz. Then, the complex analytic form of each signal is obtained using the Hilbert transform. The phase is extracted from the 10Hz signal (angle of analytic signal) and power from the 100Hz signal (amplitude of analytic signal). An uncorrected PAC measure is obtained from these two signals. To estimate the null distribution of the measure in the absence of any genuine coupling, the amplitude signal is split into two blocks at a random time point and the temporal order of those two blocks is swapped. Then, the PAC is estimated using this swapped version of amplitude and the originally extracted phase. By repeating this process and cutting at a random point, for example 200 times, we can obtain a distribution of surrogate values for which there is no genuine coupling. Finally, a corrected PAC estimate is obtained through z-score normalization of the uncorrected PAC using this distribution.

validated methods will be continually added and documented online). In the following, we denote by $x(t)$ a time-series of length $N$, $f_\phi = [f_{\phi 1}, f_{\phi 2}]$ and $f_A = [f_{A1}, f_{A2}]$ the frequency bands respectively for extracting the phase $\phi(t)$ and the amplitude $a(t)$.

**Mean Vector Length:** The Mean Vector Length (MVL) was introduced by Canolty et al. [39] and is defined as the modulus of the average complex vector formed by combining the phase and amplitude signals:

$$MVL = \frac{1}{N} \left| \sum_{k=1}^{N} a(k) e^{j\phi(k)} \right| \qquad (1)$$

Note that authors also proposed to normalize the MVL by computing surrogates using a time lag.

**Modulation Index:** Originally the Kullback-Leibler distance is used in information theory to measure dissimilarities between two probability distributions. Tort et al. 2010 [46] elegantly proposed an adaptation for measuring PAC which consists of defining a probability distribution of amplitudes as a function of phase and then comparing this distribution to a uniform one. To this end, the phase $\phi(t)$ is first cut into $n$ slices. For example, if $n = 18$, the phase is binned into 18 bins of 20° each. Then, the mean of the amplitude $a(t)$ is taken inside each bin

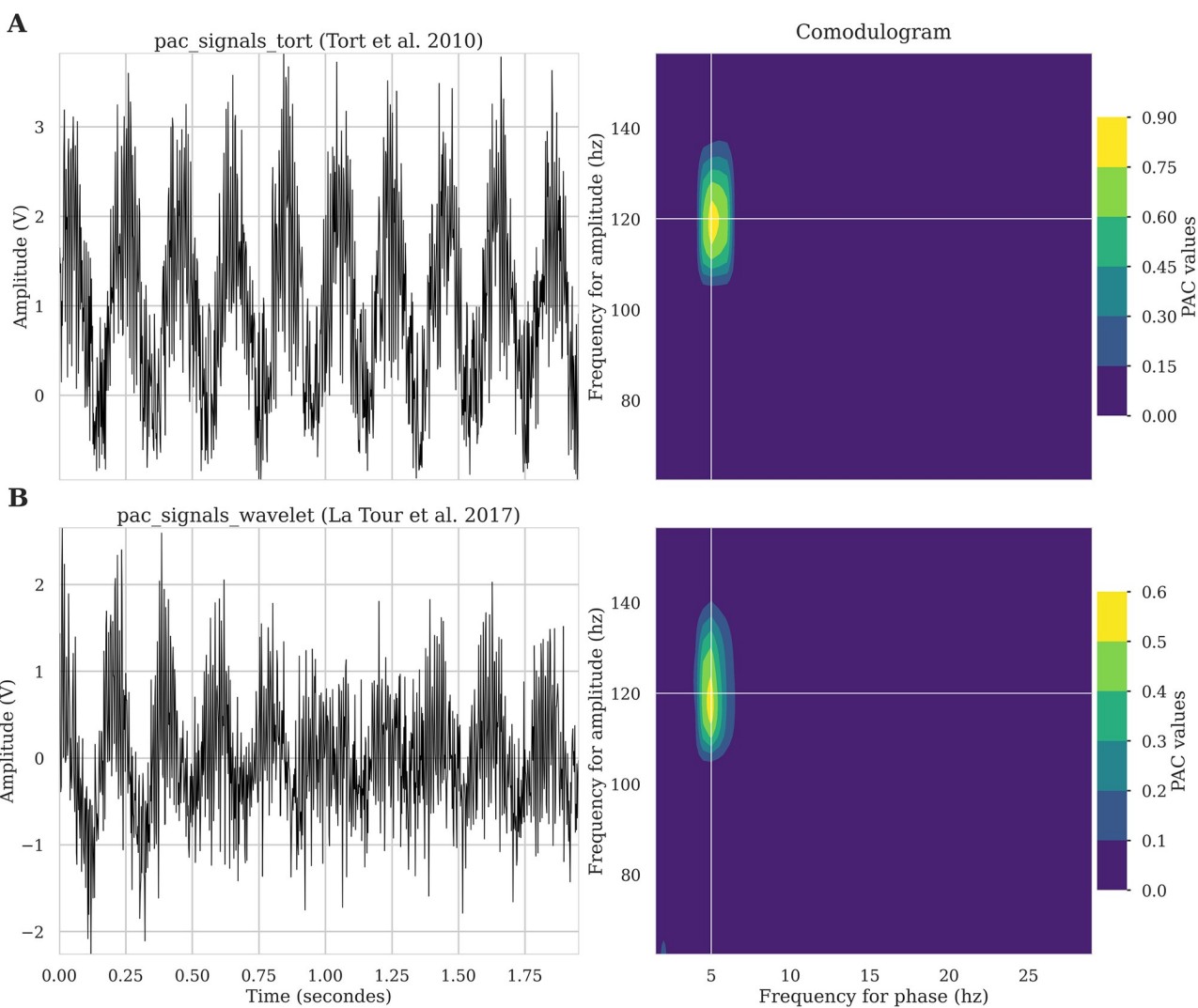

**Fig 2. Synthetic signals to simulate phase-amplitude coupling.** Illustration of synthetic signals that can be generated to simulate phase-amplitude coupling (left column) and associated comodulogram (right column). A: First row shows an example of a signal that contains a 5↔120hz coupling defined as proposed by Tort et al. [46]. B: Bottom signal also contains a 5↔20hz coupling but defined as proposed in Dupré la Tour et al. [41].

and is denoted by $\langle a(t)\rangle_{\phi}$. Through this binning operation, the phase and the amplitude are linked and can be said to be coupled. Finally, the probability distribution $P$ is obtained by dividing the amplitude inside each bin by the sum over the bins:

$$P(j) = \frac{\langle a\rangle_{\phi}(j)}{\sum_{k=1}^{n} \langle a\rangle_{\phi}(k)} \qquad (2)$$

where $\forall j \in [\![1, n]\!]$, $P(j)$ represent the normalized amplitude inside a bin. This distribution is then used to compute PAC using either the modulation index either the heights ratio.

The modulation index (MI) is obtained using Kullback-Leibler distance which measure how the probability distribution of amplitudes $P$ diverges from a uniform distribution $Q$:

$$MI = \frac{D_{KL}(P, Q)}{log(n)} \qquad (3)$$

where

$$D_{KL}(P, Q) = \sum_{k=1}^{n} P(k) log(\frac{P(k)}{Q(k)}) \tag{4}$$

Note that for a uniform distribution, $\forall k \in [\![1, n]\!]$, $Q(k) = 1/n$ and therefore the *MI* formula can be written to:

$$MI = 1 + \frac{1}{log(n)} \sum_{k=1}^{n} P(k) log(P(k)) \tag{5}$$

**Heights Ratio:** Starting from the same probability density distribution of amplitudes, the Heights Ratio (HR) [42] is defined by:

$$MI = \frac{h_{max} - h_{min}}{h_{max}} \tag{6}$$

where $h_{max}$ and $h_{min}$ are respectively the maximum and the minimum of the distribution.

**Normalized direct PAC:** The ndPac [59] is similar to the MVL with two exceptions. First, this method uses a z-scored normalized amplitude and secondly, includes a statistical test. This test uses a closed-form statistical threshold given by:

$$x_{th} = 2 \times N \times (erf^{-1}(1 - p))^{2} \tag{7}$$

With $p$ the confidence level, $N$ the number of time points and $erf^{-1}$ the inverse error function. In order to find this threshold, the amplitude is assumed to be normally distributed (zero mean and unit variance) and the phase is assumed to be uniformly distributed between $-\pi$ and $\pi$. To this end, the mean and deviation across time points of the amplitude are used to perform a z-score normalization in order to approximate the original assumptions. Finally, every value of coupling exceeding this threshold is considered as reliable, at a given confidence level. Otherwise, if the value of coupling is below this threshold it is set to zero.

**Phase-Locking Value:** The phase-locking value (PLV) [45, 60] looks only at the phase consistency across trials. The PLV consists of extracting the phase of the amplitude $\phi_a$, subtracting it from the phase of slower oscillations, projecting the resultant time series into the complex circle and finally, calculating the mean of the length vector:

$$PLV = \frac{1}{N} |\sum_{k=1}^{N} e^{j(\phi(k) - \phi_a(k))}| \tag{8}$$

**Gaussian-Copula PAC:** Mutual Information (MI) can be used to quantify pairwise statistical dependence between many different types of variables on a common and meaningful effect size scale (bits) [61]. However, MI can be difficult to estimate in practice as it requires sampling the full joint distribution of the two considered variables, each of which can themselves be multivariate. Gaussian-Copula Mutual Information (GCMI) is a recently proposed semi-parametric estimation technique [62] which has some advantages for estimating MI from neural data. GCMI exploits the fact that mutual information is copula entropy: MI does not depend on the marginal distributions of the variables, but only on the copula function which describes their dependence. GCMI therefore first transforms the inputs to be standard normal. This copula-normalisation step involves calculating the inverse standard normal cumulative density function (CDF) value of the empirical CDF value of each sample, separately for each input dimension, before using a parametric, bias-corrected, Gaussian MI estimator. This provides a lower bound estimate of the MI, without making any assumption on the marginal

distributions of the input variables. GCMI is rank-based, robust and scales well with multidimensional variables as the Gaussian model reduces the curse of dimensionality that faces other methods such as those involving binning.

In the case of PAC, the phase $\phi(t)$ and the amplitude $a(t)$ are still extracted using the Hilbert transform or using wavelets. The rank normalization is first applied on the amplitude $a(t)$. To preserve the cyclic nature of phase, it is represented as points on the unit circle of the complex plane, represented as a 2d variable for GCMI. This 2d variable is built by concatenating the sine and cosine of the phase $\phi(t)$, $[sin(\phi(t)), cos(\phi(t))]$ projecting it into the unit circle (or alternatively by normalizing away the amplitude of the complex value). The copula normalization procedure is then applied to each dimension of this 2d variable. PAC is measured as the GCMI between the copula-normalized 1d high-frequency amplitude and the 2d representation of low-frequency phase:

$$gcPAC = I(a(t); [sin(\phi(t)), cos(\phi(t))]) \tag{9}$$

Since the gcPAC is rank-based, it is, by construction, invariant to overall amplitude shifts which means that a power increase does not lead to an increase of coupling. Note that measuring PAC using mutual-information has already been proposed using nearest-neighbour estimators [43]. Similarly to the other measures, Tensorpac provides a tensor-based implementation of the GCMI computed between two continuous variables.

While there is still no gold standard for choosing the most appropriate PAC method, some of them are by construction less suitables for measuring genuine coupling. In particular those for which an increase of power would also lead to an increase of coupling (amplitude dependency). Therefore, we recommend choosing methods like the MI, HR or gcPAC which are all not affected by the magnitude of amplitude. The main PAC methods implemented in Tensorpac are presented in Fig 3.

**Statistical analysis of PAC.**   The absence of PAC in a signal could be related to several parameters that have been previously described [45]. Each one of the proposed PAC methodologies presents some advantages or limitations and may not be appropriate for all types of analysis. These methods exhibit differences in terms of robustness to noise, as well as modulation width, neither of which are necessarily amplitude independent [46]. In addition, PAC estimations may be biased due to limited amounts of data being available.

Generally, these limitations can be taken into account by computing a surrogate null distribution and using this to correct or normalize the PAC measure. To this end, several methods exist, all based on a common idea: introducing a small change into the data such that the temporal characteristics of the time-series are preserved but the relationship between the phase and the amplitude is disrupted. Among existing methods, [39] introduced a time lag to the amplitude, while [46] swap amplitude and phase trials and [57] swap time blocks, cut at a random point. The latter method, with only two blocks, has been described as the most conservative strategy to generate the distribution of PAC that can be observed by chance [48]. Finally, this null distribution is then used to perform non-parametric inference or to correct the measure estimated from the data (usually by subtracting the mean and divide by the deviation of this distribution). An example of a corrected PAC is presented in Fig 4.

Finally, the null distribution can also be used in order to infer the p-value. Indeed, the p-value is defined as the proportion of surrogates that are exceeded by the true value of coupling. For a comodulogram that contains several phase and amplitude frequencies, Tensorpac also includes a correction for multiple-comparisons. By default, it uses the maximum statistics which provides a threshold that controls family-wise error rate [63, 64]. The procedure

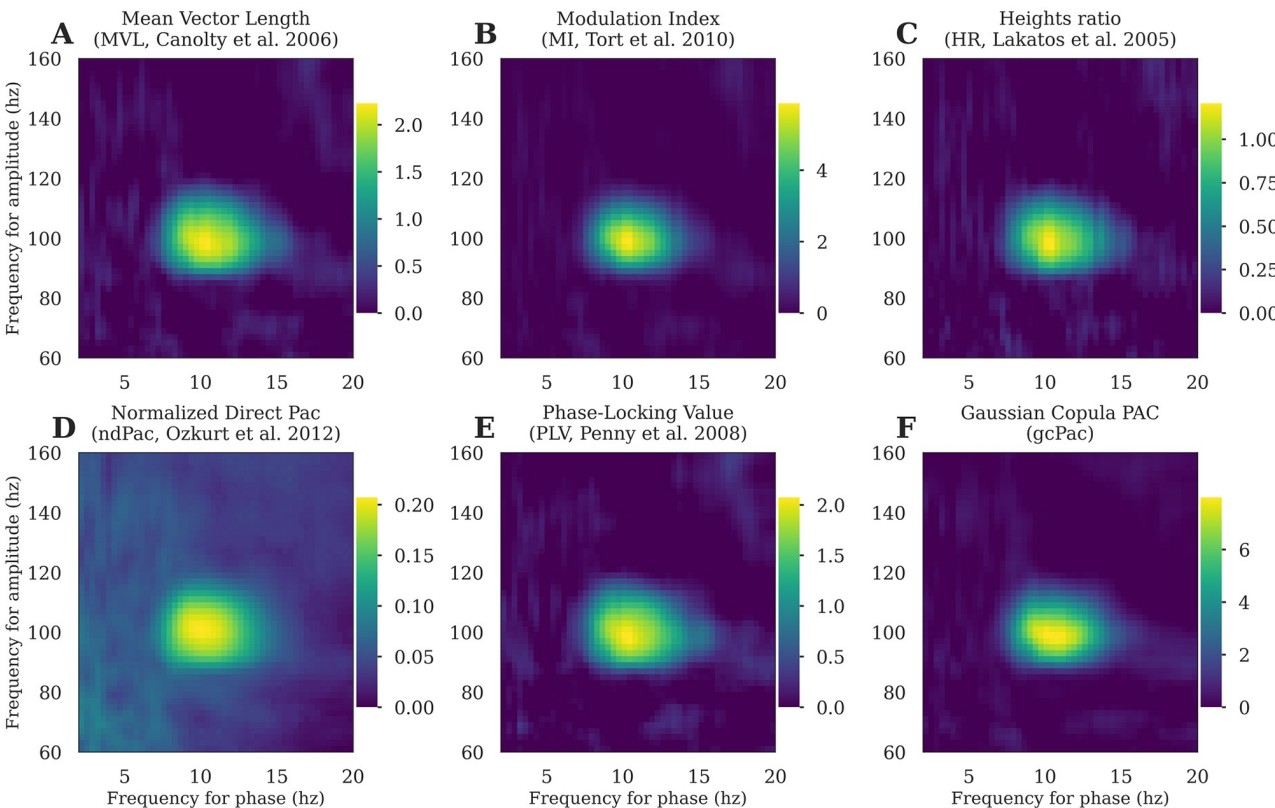

**Fig 3. Comparison of the main PAC methods currently implemented in Tensorpac.** The comodulograms were computed using the A: MVL, B: MI, C: HR, D: ndPac, E: PLV and F: gcPAC, from 20 trials of simulated data containing a 10↔100hz phase-amplitude coupling. The data is available in Tensorpac and can be used to validate and benchmark other methods.

consists in taking the maximum of the surrogates over all of the computed phases and amplitudes and using this distribution of maxima to infer the p-value.

**Event-related phase-amplitude coupling (ERPAC).** One issue that has been raised is that, since PAC is computed across time, non-stationary signals can cause the appearance of

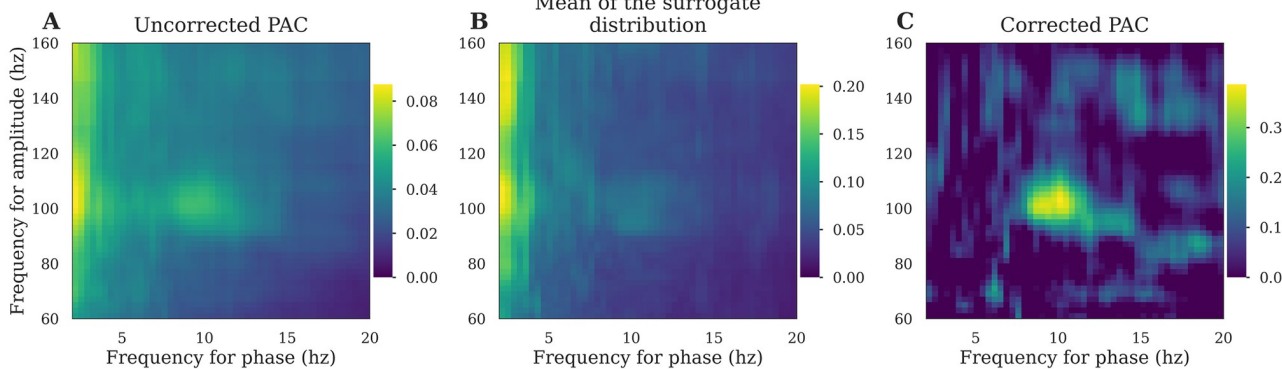

**Fig 4. Comparison between corrected and uncorrected PAC.** A: PAC comodulogram is computed for several (phase, amplitude) pairs. B: For each of those pairs, we estimate the distribution of surrogates and plot the mean comodulogram of these permutations. Note that both uncorrected and surrogate PAC comodulograms exhibit a spurious peak in the very low frequency phase. C: The true 10↔100hz coupling is finally retrieved by subtracting the mean of the surrogate distribution (panel B) from the uncorrected PAC (panel A).

spurious coupling [48]. An illustrative example taken from the same study explains that if there is an induced phase locking of lower frequencies and simultaneously a high frequency power increase, a coupling between them is going to be observed. Interestingly, a complementary approach to the time-averaged PAC has been proposed and consists of computing time-resolved PAC across trials [47]. Accordingly, the Event-Related PAC (ERPAC) measure is based on a circular-linear correlation [65] which evaluates the Pearson correlation, across trials, of the amplitude $a_t$ and with the sine and cosine of the phase $\phi_t$. We denote by $c(x, y)$ the Pearson correlation between two variables $x$ and $y$, $r_{sx} = c(sin(\phi_t), a_t)$, $r_{cx} = c(cos(\phi_t), a_t)$ and $r_{sc} = c(sin(\phi_t), cos(\phi_t))$ hence, the circular-linear correlation $\rho_{cl}$ is defined as:

$$\rho_{cl} = \sqrt{\frac{r_{sx}^2 + r_{cx}^2 - 2r_{sx}r_{cx}r_{sc}}{1 - r_{sc}^2}} \tag{10}$$

In contrast to the original Matlab version [47], we implemented a tensor-based version of the ERPAC (*tensorpac.EventRelatedPac*). Similarly to the gcPAC, we also introduce the Gaussian-Copula Event-Related PAC which this time measures the information shared by the phase and the amplitude across trials, at each time point illustrated in the Fig 5. It is noteworthy that a measure of event-related PAC has been proposed using a mutual information framework [43]. In addition, it has also been proposed to compute PAC using the power spectrum on sliding windows [66].

## Additional cross-frequency tools

**Distribution of amplitudes and preferred phase.** The preferred-phase (PP) is defined as the phase for which the distribution of amplitudes is maximum. This can be used to find out if amplitudes are aligned at a specific phase angle [28]. To compute the PP (*tensorpac.PreferredPhase*), the probability density distribution of amplitudes is first generated according to a number of phase slices (just as the modulation index and heights ratio). Then, the phase bin for which the amplitude is maximum is defined as the preferred phase. Usually, the preferred

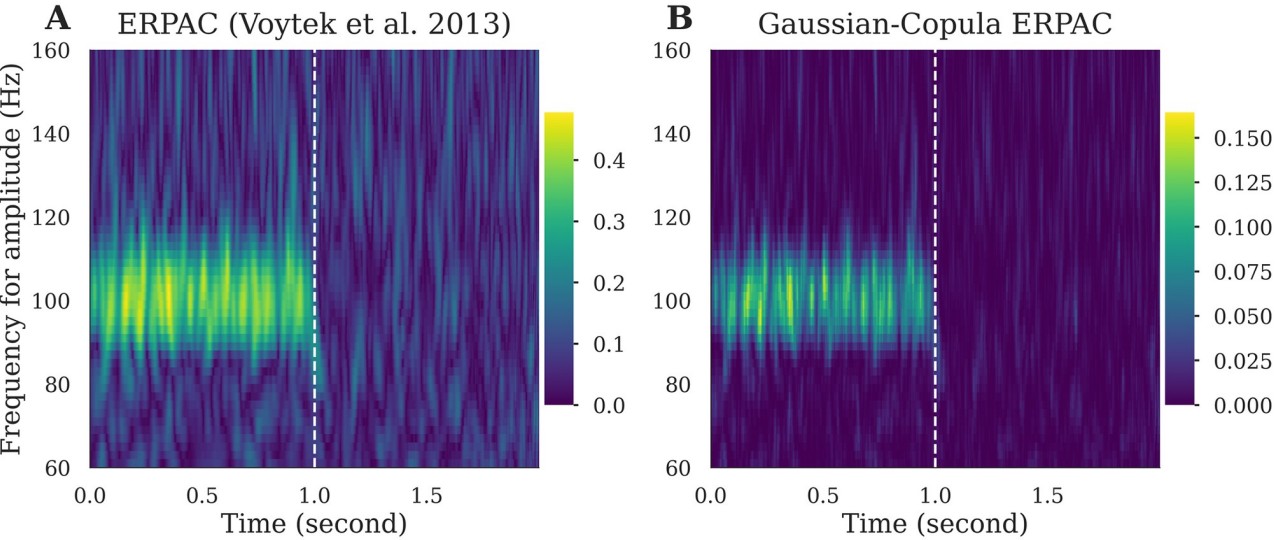

**Fig 5. Example of event-related phase-amplitude coupling (ERPAC).** We first generate 300 one-second trials each containing a 10↔100hz coupling. Next, one-second of random noise is appended to these signals. The depicted A: ERPAC and B: gcERPAC represents time-resolved PAC estimation over the two-second window, computed with a phase between [9, 11] hz and for multiple amplitudes.

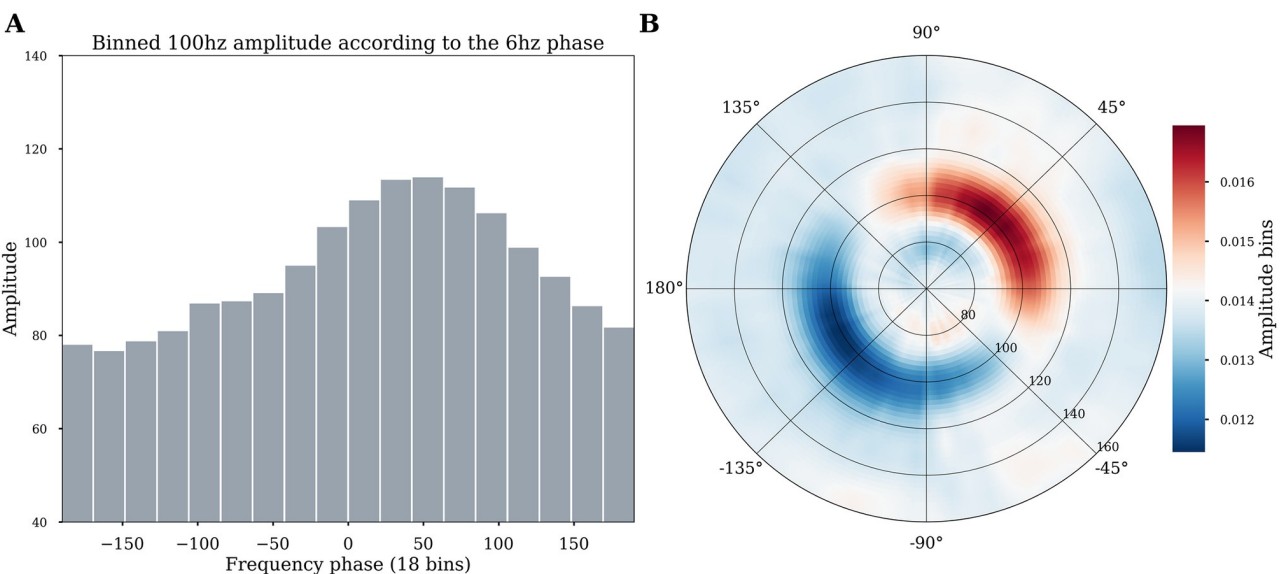

**Fig 6. Estimation of the preferred phase.** Illustrative example of the preferred phase estimation based on 100 trials generated with coupling between 6hz↔100hz and where the amplitude is locked to the 6hz phase at 45˚ ($\pi/4$). A: The 100 hz amplitude is first binned according to the phase using 18 slices of 20˚ each. The sum of the amplitude inside each slice is plotted as a histogram and the preferred phase is identified as the phase for which the amplitude is maximum (45˚). B: An alternative polar visualization available in Tensorpac displays the strength across multiple amplitude frequency bands. Phase is binned as before, but now multiple amplitude signals from different bands are calculated for each phase bin. In these polar plots, the angle represents the phase of the low-frequency (here 6Hz), and the radial axis represents different frequencies considered for the amplitude signal. The color depicts the average value of the amplitude of a given frequency inside the corresponding phase bin. The preferred 45˚ phase for the 6hz↔100hz PAC is clear in this representation.

phase is reported using a histogram (see Fig 6A), where a specific phase and a specific amplitude band have been used. Here, we introduce a new plotting method where, still for a single phase, but now binned amplitudes in consecutive frequency bands can be observed using a polar representation (Fig 6B). This provides a more fine-grained representation where the preferred phase can be observed with a wide range of amplitude frequencies. Both methods and visualizations are available in Tensorpac (*tensorpac.PreferredPhase.pacplot* and *tensorpac.PreferredPhase.polar*).

**Phase / amplitude frequency interval optimization.**   When choosing the parameters to use in PAC analysis, researchers are often confronted with important decisions related to parameter selection. Even if the frequency bands for phase and amplitude are chosen based on a scientific hypothesis or previous reports in the literature, it is often impossible to know what the optimal frequency intervals are in the data one is analyzing. One might argue that it makes little sense to compute theta-gamma coupling in canonical frequency bands for example using 4-7 Hz (theta phase) and 30-70 Hz (gamma amplitude) if these bands don't really capture key oscillatory modulations in the data at hand. We therefore reasoned that it would be useful to (a) be able to check for the presence of peaks in the power spectrum to potentially guide the choice of the bandwidth for filtering [48], and (b) to automatically search for the best frequency intervals in a data-driven manner. Tensorpac provides functionalities that can help address these issues. First of all, a standard tool to compute the Power Spectrum Density (PSD) is available and adapted for standard electrophysiological data formats, i.e. datasets organized as an array with the number of epochs as rows and the number of time points as columns (See example in Fig 7A). This can be used to identify prominent peaks either for the low or high frequency components. More importantly, in order to address the question of how to optimize the selection of the starting and ending frequencies for the intervals to use for PAC

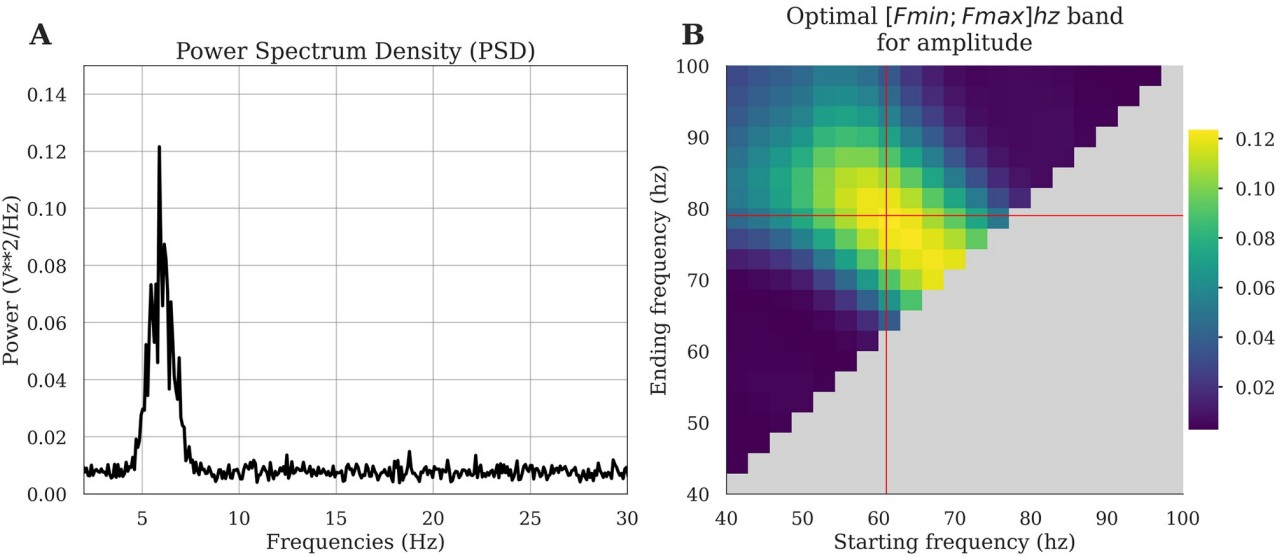

**Fig 7. Investigation of the presence of a phase peak and data-driven exploration of the optimal bandwidth.** In this example, we first generated a 6↔70hz phase-amplitude coupling. A: The PSD retrieves the presence of the phase peak around 6hz. B: For a fixed phase filtered in [5, 7] hz, we search for the optimal amplitude band, defined as the bandwidth for which the PAC is maximum. The triangular freq-freq representation depicts coupling strength across many possible combinations of amplitude frequency bounds, where the x-axis corresponds to the starting frequency and the y-axis to the ending frequency. Here, the PAC is maximum for an amplitude range of [61, 79] hz.

computation, Tensorpac allows for the possibility of defining triangular vectors and computing PAC for a range of starting and ending frequencies ($PAC(F_{min}, F_{max})$). In addition, this triangular search can be used to find the best interval for the amplitude or for the phase. The visualization of the results (*tensorpac.Pac.triplot*) can be used to determine the hotspots within a triangular representation of PAC, i.e. determining the Fmin/Fmax combination that corresponds to PAC peak (see Fig 7B): The x-axis determines the starting frequency ($F_{min}$) and the y-axis the ending frequency ($F_{max}$). The maximum coupling that emerges from this triangular representation can be taken as an indication for the optimal frequency interval to use [$F_{min}$, $F_{max}$]. Note here that the standard comodulogram often used in PAC analyses is obtained by computing PAC in successive (phase, amplitude) pairs of bands with pre-defined bandwidths. While useful for identifying coupling the comodulogram is not suitable for identifying the optimal starting and ending frequencies to use. Taken together, the PSD tool and this exhaustive Fmin/Fmax search for best frequency bounds are valuable tools that can guide decisions regarding parameter selection in PAC analyses.

As a side note, the choice of the filter bandwidth for the phase and amplitude is still debated. While some previous studies recommended filtering the amplitude with a bandwidth twice as large as the one used for phase (2:1 ratio) [48], a recent study suggests that a 1:1 ratio might be better as this could prevent smearing [67].

**Statistical test of stationarity.** As mentioned earlier, stationary signals are a prerequisite for calculating phase-amplitude coupling. Tensorpac includes a function (*tensorpac.stats.test_-stationarity*) to perform an Augmented Dickey-Fuller test (ADF) [68]] to test the stationarity of time-series. The null hypothesis of the ADF test is that there is a unit root in the time-series. Said differently, H0 represents a non-stationary signal. Tensorpac uses the Statsmodels Python package [69] and returns a dataframe that contains, for each epoch, the p-values, a boolean if H0 has been accepted or rejected, the statistical test and critical values at 0.05 and 0.01.

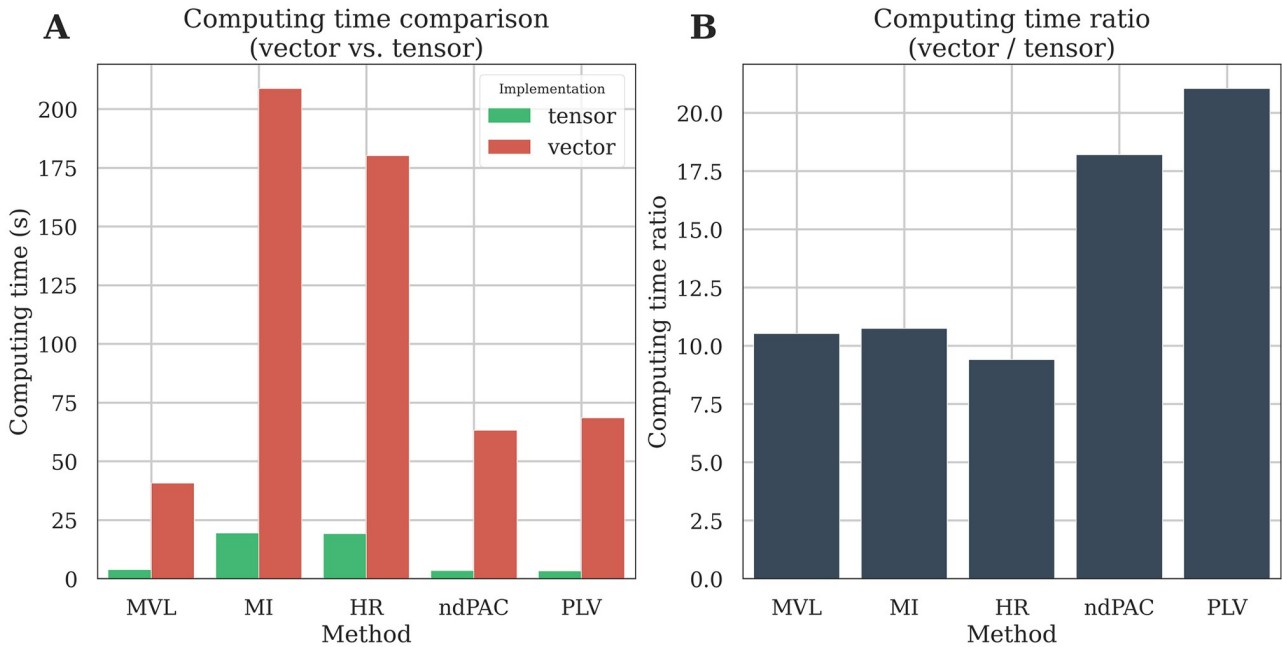

**Fig 8. Computation time of the vector and tensor-based implementations.** We first generate a relatively small dataset composed of 100 trials of 3000 time points each. We then evaluated the comodulogram by extracting 26 phases and 24 amplitudes. The comodulogram is then either computed using one-dimensional time-series (vector-based) or directly using multidimensional arrays (tensor-based). Computing time is compared as a function of PAC method (A) or as a ratio where the computing time using tensors is divided by the one using vectors (B).

## Tensor-based implementation

Traditionally, phase-amplitude coupling measures are implemented in a vector fashion where only a single time-series is processed at a time. This is the most straightforward implementation and the easiest to read, however computing PAC on relatively large datasets can be dramatically slow, especially when the number of dimensions increases (i.e the number of trials, phases and amplitudes). In Tensorpac, all of the implemented PAC methods have been adapted in order to support multidimensional computations and thereby decrease computing time. This was made possible by using the NumPy function einsum which uses the Einstein summation convention in order to perform linear algebraic array operations. In order to illustrate the gains in terms of computing time, we compared the vector-based and the tensor-based implementations on simulated data (100 trials and 3000 time points). We then computed a comodulogram with 26 phases and 24 amplitudes. Fig 8 presents the comparison of computing time per method (Fig 8A) and the ratio (Fig 8B). The tensor-based implementation is between 6 times to more than 12 times faster than the vector-based one depending on the PAC method that is used. This gain in terms of computation time would be obviously even larger when considering an increasing number of phases and amplitudes or if the permutations for assessing statistics have to be measured.

## The case of spurious cross-frequency coupling

Previous research has established that non-oscillations, non-stationary signals and sharp waveforms (e.g the mu rhythm in sensorimotor areas) can all produce spurious PAC even in the absence of true phase-amplitude coupling [48, 52, 70, 71]. While considering non-oscillations is an important aspect of brain data analysis [50, 72], the fact that it can also produce non-existent couplings is problematic for future work on cross-frequency coupling. Most of the

standard PAC methods rely on an instantaneous estimation of the phase and amplitude using Hilbert transform or Morlet wavelets. However, both approaches can introduce spurious PAC from sharp waveform features. A few methods, not based on instantaneous estimations of phase and amplitude, have been introduced to tackle the issue of spurious PAC. These include for example auto-regressive models [41] and bispectral approaches [73]. The default instantaneous signal decompositions in phase and amplitude included in Tensorpac at the time of publication do not solve the ongoing debate on spurious versus genuine PAC, nevertheless the package provides a robust and convenient framework for the incorporation of additional PAC methods which may be developed to address this issue in the future.

More generally, when using Hilbert or wavelets, several guidelines and precautions have been proposed in order to limit spurious PAC and control the accuracy of an estimation [48]. First, a visual inspection of the time-series could be performed in order to exclude non-stationary signals. In addition, a clear peak at the phase frequency should be visible in the PSD to warrant further exploration of PAC (this can be examined in Tensorpac using *tensorpac.utils. PSD*). Still, the Fourier transform decomposes a signal in a sum of sines and cosines which means that it can only partially capture non-oscillatory features [70, 72]. Instead, one might consider using a matching pursuit algorithm or an empirical mode decomposition since both do not make the assumption of a sinusoidal basis [72]. In addition to these two alternatives, a method based on cycle-by-cycle analysis has recently been proposed in order to extract temporal features [72]. In principle, if a peak in the PSD is observed at the frequency of the phase, the width of this peak should be used to then extract the instantaneous phase. When filtering the amplitude, the width of the frequency band should be large enough to contain the side bands of the lower frequency band. In practice, it means that if the band used for low frequency phase is [8, 10]Hz, the bandwidth of the amplitude should be at least 20Hz. The presence of coupling can also be further explored by binning the amplitude according to phase slices (*tensorpac.utils.BinAmplitude*). In absence of coupling the distribution should be uniform while in presence of coupling, this distribution of amplitude should be closer to a normal distribution. Tensorpac also includes the possibility to realign time-frequency representations based on a phase peak (*tensorpac.utils.PeakLockedTF*). If a coupling exists, a rhythmic pattern of higher frequency (e.g. gamma) amplitude bursts should be observed.

## Documentation and API provision

Tensorpac is a Python 3 package and is distributed under a BSD-3-Clause license. This package relies on NumPy [74], SciPy [75], Joblib for parallel computing and Matplotlib for plotting. In addition, Pandas [76] and Statsmodels [69] are required if the stationarity test is to be performed. We also provide full documentation for the package (https://etiennecmb.github.io/tensorpac) which is automatically built using sphinx. This documentation also explains how to install Tensorpac. An API tab is accessible from this documentation and describes the most up-to-date implemented functions and descriptions (using NumPy doc convention). It also features a gallery of examples, built with sphinx-gallery, which demonstrate the use of Tensorpac's main classes and functions and a step-by-step tutorial based on real intracranial EEG data recorded during a center-out motor task [23]. For beginners or non-python users that want to cross the open-source bridge, we added a public Gitter chat for answering questions. Finally, the code follows PEP8 and Flake8 guidelines for code readability. We also added a suite of unit-tests (smoke tests and functionnal tests) that are systematically launched on Linux and Windows systems. Finally, Tensorpac functionalities can easily be combined with other open-source brain data analysis and visualization tools developed by our group including visbrain [77, 78] and Neuropycon [79].

## Availability and future directions

This paper introduces the workflow and functionalities of Tensorpac, a free and open-source Python toolbox with a tensor-based implementation of both time-averaged and trial-averaged phase-amplitude coupling measures. In addition to those most frequently used methods we also presented some unique features such as the preferred phase or an exhaustive research of frequency bounds. As spurious coupling can be observed in many scenarios, we also provide additional tools and statistics to control both the reliability of an estimation. The latest version of Tensorpac is hosted on Github (https://github.com/EtienneCmb/tensorpac) but can also be installed via the pip command from a regular terminal. Tensorpac comes with an online documentation that describes installation options, also for contributors, the functionalities and illustrative examples. We plan to continue adding new methods to the Tensorpac toolbox and we encourage collaborative development and contributions from the community. In particular, we welcome contributions of new methods of PAC estimation but also tools that can help control the reliability of PAC metrics and reduce spurious detections.

## Financial disclosure statement

EC and AB were supported by the French National Agency (ANR-18-CE28-0016-01)). http://anr.fr.

EC was also supported by funding via a Natural Sciences and Engineering Research Council of Canada (NSERC). https://www.nserc-crsng.gc.ca/.

JLPS acknowledge support from the Brazilian Ministry of Education (CAPES grant 1719-04-1) and the Fulbright Commission to J.L.P. Soto.

https://www.iie.org/programs/capes.

KJ was supported by funding from the Canada Research Chairs program and a Discovery Grant (RGPIN-2015-04854) from NSERC (Canada), a New Investigators Award from FQNT (2018-NC-206005) and an IVADO-Apogée fundamental research project grant. This research is also supported in part by the FRQNT Strategic Clusters Program (2020-RS4-265502—Centre UNIQUE—Union Neurosciences & Artificial Intelligence—Quebec).

http://www.frqnt.gouv.qc.ca/.

https://ivado.ca/.

TC acknowledges support through the Centre de Recherches Mathématiques (CRM).

RAAI was supported by the Wellcome Trust [214120/Z/18/Z].

https://wellcome.ac.uk/.

The funders had no role in study design, data collection and analysis, decision to publish, or preparation of the manuscript

## Author Contributions

**Conceptualization:** Etienne Combrisson, Andrea Brovelli, Robin A. A. Ince, Karim Jerbi.

**Data curation:** Etienne Combrisson, Timothy Nest.

**Formal analysis:** Etienne Combrisson, Andrea Brovelli, Robin A. A. Ince.

**Funding acquisition:** Aymeric Guillot, Karim Jerbi.

**Investigation:** Etienne Combrisson.

**Methodology:** Etienne Combrisson, Robin A. A. Ince, Juan L. P. Soto.

**Project administration:** Etienne Combrisson.

**Resources:** Etienne Combrisson, Karim Jerbi.

**Software:** Etienne Combrisson, Timothy Nest, Juan L. P. Soto.

**Supervision:** Aymeric Guillot, Karim Jerbi.

**Validation:** Etienne Combrisson, Timothy Nest.

**Visualization:** Etienne Combrisson, Karim Jerbi.

**Writing – original draft:** Etienne Combrisson, Timothy Nest, Andrea Brovelli, Aymeric Guillot, Karim Jerbi.

**Writing – review & editing:** Etienne Combrisson, Timothy Nest, Andrea Brovelli, Robin A. A. Ince, Aymeric Guillot, Karim Jerbi.

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
