## [Decision Letter · Decision Letter 0]

25 Jun 2020

Dear Dr. Combrisson,

Thank you very much for submitting your manuscript "Tensorpac : an open-source Python toolbox for tensor-based Phase-Amplitude Coupling measurement in electrophysiological brain signals" for consideration at PLOS Computational Biology.

As with all papers reviewed by the journal, your manuscript was reviewed by members of the editorial board and by several independent reviewers. In light of the reviews (below this email), we would like to invite the resubmission of a significantly-revised version that takes into account the reviewers' comments.

We cannot make any decision about publication until we have seen the revised manuscript and your response to the reviewers' comments. Your revised manuscript is also likely to be sent to reviewers for further evaluation.

Sincerely,

Dina Schneidman-Duhovny

Software Editor

PLOS Computational Biology

Dina Schneidman-Duhovny

Software Editor

PLOS Computational Biology

Reviewer's Responses to Questions

**Comments to the Authors:**

Reviewer #1: This paper describes a Python based toolbox to estimate, most of all, phase-amplitude

coupling. This area of research has recently attracted much interest and

therefore the toolbox is potentually useful for a large number of researchers.

The paper is very well written and, perhaps with one exception to be discussed

below, free of errors. In addition to implementing existing methods,

the authors developed a new visualization technique shown in Fig.6B which

is really nice. I still have some objections which should be addressed by the authors.

Major

1. The authors present a formula to estimate a treshold written as 2 × ((1 − )^−1)^2.

For a z-scored variable, the threshold would be the square root of that. Perhaps it is meant

to be the threshold for the square of the z-scored MVL value, but here the authors should

be more specific of what exactly is thresholded. Also, writing the exponent -1 in the formula

after the brackets for the argument of erf is confusing. That looks like 1/erf(1-p). Finally, to my opinion this z-scoring and the statistical interpretion as a Gaussian distribution with zero mean and

unit standard deviation doesn't make sense. The distribution of MLV under the null hypothesis

isn't even remotely close to a Gaussian distribution. I don't blame the authors. They just implemented

methods proposed by others, but a warning remark would be helpful.

2. The corrected PAC shown in Fig.4C is constructed by subtracting the mean and dividing by

the standard deviation of the surrogates, right? At least, this is how understand

the sentence "... usually by subtracting the mean and divide by the deviation of this distribution" on p.13. But why are the results so small? Not any of the shown results would survive the threshold discussed above. As stated above, I doubt that the threshold is based on accurate statistics, but I would still expect values in the order of 1 even when the null hypothesis is true, and I would expect very high values for real effects like in Fig.3 ndPAC. If it is not normalized here, why does the difference between Fig.4.A and Fig.4.B has values larger than 0.3 in the central peak? Finally, I also wonder whether MVL is interpretable without any normalization because the results will depend on overall signal amplitude.

Details:

1. As far as I know, what is called PLV in this paper, is usually called ITC (inter trial coherence)

and what is called PS here I would call PLV. Perhaps terminology varies. Just check.

2. Last equation of section "Kullback-Leibler distance:"

Brackets are missing. It should read log(P(k)) and not log(PK).

3. p.11 "This distribution is then

used to compute PAC using either the KLD." Or the normalized MVL?

ent peaks"

Reviewer #2: Uploaded as an attachment

Reviewer #3: == SUMMARY ==

The authors present an open-source toolbox ‘tensorpac’ for calculation of phase-amplitude coupling (PAC) measures. The toolbox features large speed gains in PAC-computation due to the usage of tensors in contrast to vector-based computation. Given clarifications regarding advantages with respect other existing toolboxes for PAC and statistical analysis, as well as extending the documentation, I believe this a valuable methodological contribution.

== DETAILS ==

The authors list 4 points as main advantages of their toolbox:

(1) higher computational efficiency because of tensor computations and parallelization

EVIDENCE: In Fig. 8, large gains in computation time are seen. Since at the core, the gains are achieved by relying on the Einstein summation representation, it would be interesting to expand on this in section 3.

It is neat that the scripts producing the figures are included and the figures can all be reproduced for run-time comparisons. The authors list pacpy (which seems not actively supported anymore) and pactools as comparisons to their toolbox. As pactools also offers parallelization, it is of interest to compare potential speed gains directly to pactools, not only vector-based usage in a loop of tensorpac functions.

(2) the implementation of all most widely used PAC methods in one package

EVIDENCE: According to documentation of pactools, there also numerous methods to calculate the comodulogram are implemented, even more than tensorpac, e.g. driven auto-regressive methods and bispectra, which makes the statement that other packages offer a ’limited choice of PAC computations’ not backed up. A more elaborate discussion of the criteria for selecting the chosen methods implemented in ‘tensorpac’ and highlighting improvements over existing packages would be helpful.

The authors stay agnostic to which metric should be used, but possibly there are cases for which each metric has advantages (briefly alluded to in section 1.4) and the brief descriptions of the metrics could be expanded in that respect.

From the API documentation, there is also a Gaussian Copula Mutual information PAC/ERP-PAC function, but it is not discussed in the paper, maybe it would be of interest to include in the manuscript.

(3) advantages for the statistical analysis of PAC measures

EVIDENCE: the authors provide methods for computing signal stationarity and surrogate null distributions. An often-desired outcome of PAC analysis is to derive a significance level of obtained comodulogram values, disentangling genuine PAC from spurious PAC. It would be helpful to provide discussion of it in the article (some functions for inferring p-values are present in the toolbox), as well as treatment of multiple comparison correction (cluster-based?) and overfitting & hyperparameter selection, for instance in the context of the proposed triangular search for optimizing frequency intervals.

(4) extended PAC visualization capabilities.

EVIDENCE: the toolbox provides visualization of comodulograms, frequency-amplitude histograms and polar plots, which are easy to produce and helpful visualizations. As a user, I would find peak/trough-locked (to slow oscillation) time domain & time-frequency plots useful in addition.

Minor

------

- the ‘<->’ arrow could be substituted by a proper arrow for increased readability

- there is some inconsistency in referring to KL-divergence. ‘KL-distance’ ‘distance of Kullback-Leibler’

- in the context of PAC, changing the labels/subsections from KLD to Modulation index could be considered for familarity

- p.6 and following, sometimes there is no space after lots of formulas ‘MIformula’ ‘amplitudes Pdiverges’

- sometimes an abbreviation is introduced, only to be used once or twice (e.g. PLV, PS, etc) -> use full name throughout for better readability?

- Fig.3: I would recommend using the full name of the measure in the subplot titles. Also, here it is noticeable that the scale of the measures is different, maybe the authors could comment on boundedness or ranges of measures in the description of the measures

- figure labels and legends are small at times

- p.6 ‘Height-Ration’ -> Height-Ratio

== SOFTWARE & WEBSITE ==

- the toolbox has a clearly indicated license (BSD-3) and I was able to install the toolbox via pip without errors, dependencies are clearly stated.

- statement of need: The high-level description on the documentation page could be expanded to highlight the speed gains by using tensor so they are clearer at first sight. (I think it is a nice selling point)

- tutorial link: at the moment there is only a GIF ‘under construction’... it would be good to have just a simple example placed there, maybe the one from the nice Fig.1 illustration? Or delete this link and have only the examples page?

- examples: The examples include nice illustrations, but the high-level text could be extended to clarify the purpose of the example, e.g. for “Compute p-values” the text only says: ‘For the visualization, we used a comodulogram.’, for ‘Compare surrogate methods’ -> ‘Surrogates are used to generate a chance ditribution in order to correct the PAC estimation.’. Extension to something like ‘This example illustrates how …’ would be informative for users, especially novice.

- the API documentation is too brief. For instance, the documentation for the ‘ndpac’ method just reads “Normalized direct Pac.“ or ’swap_pha_amp’ : “Swap phase / amplitude trials.” It would be good to include a brief rationale for each function that would clarify what happens to the input.

- the main PAC methods have very short function names, possible it could enhance readability giving them slightly more descriptive names? e.g.: hr -> heights_ratio, ps -> phase_synchrony etc

- some of the function names are in camel case (BinAmplitude) and some have snake_case, unification would be good.

- testing: the tests currently included test basic input output relations (smoke tests), but it would be helpful to include functional tests of PAC computation for synthetic signals, for which the ground truth is known.

- paper figures scripts: I needed to adjust the path in other to reproduce the figures from the paper in order for the script not to crash; the path could be changed to a relative one, so this does not happen

- reference for mvl method is cut short in the API doc

**Have all data underlying the figures and results presented in the manuscript been provided?**

Reviewer #1: Yes

Reviewer #2: Yes

Reviewer #3: Yes

PLOS authors have the option to publish the peer review history of their article (what does this mean?). If published, this will include your full peer review and any attached files.

Reviewer #1: **Yes: **Guido Nolte

Reviewer #2: No

Reviewer #3: No
---

## [Decision Letter · Decision Letter 1]

2 Sep 2020

Dear Dr. Combrisson,

We are pleased to inform you that your manuscript 'Tensorpac : an open-source Python toolbox for tensor-based Phase-Amplitude Coupling measurement in electrophysiological brain signals' has been provisionally accepted for publication in PLOS Computational Biology.

Best regards,

Dina Schneidman

Software Editor

PLOS Computational Biology

Reviewer's Responses to Questions

**Comments to the Authors:**

Reviewer #1: The authors have addressed by concerns.

Reviewer #2: Combrisson et al. have provided a thorough revision of the manuscript and successfully addressed all of my concerns with much detail. In particular, the additional text concerning the interpretation of measured PAC is well written and is effective in highlighting the utility of specific Tensorpac methods in that regard. I also appreciate that the authors provided new tutorials with real iEEG data, which will help new users navigate the process of performing PAC analyses on their data. The updating of code for the generation of the main figures is also very useful.

As for the issue with integration of Numba/Cython into the code pipeline, it is helpful to see that this decreased the computational time of several of the functionalities -- especially since the speed of Tensorpac is a key selling point. However, I also agree with the authors that several of the issues of combining JobLib and Numba/Cython still have not been worked out by the developers. As the authors suggest, they could consider this integration as a longitudinal endeavor as the code libraries are further developed, potentially leading to further increases in computational efficiency of Tensorpac in the future.

Overall, I am enthusiastic about the publication of this work, and in my opinion, Tensorpac represents a valuable contribution to the field of computational and systems neuroscience.

Reviewer #3: I thank the authors for the reply to my comments, which addressed all my concerns. I also enjoyed reading the thoughtful replies to the comments of other reviewers & the inclusion of the paragraph about spurious PAC & inclusion of a real data example tutorial.

I now recommend this article for publication.

**Have all data underlying the figures and results presented in the manuscript been provided?**

Reviewer #1: Yes

Reviewer #2: Yes

Reviewer #3: Yes

PLOS authors have the option to publish the peer review history of their article (what does this mean?). If published, this will include your full peer review and any attached files.

Reviewer #1: **Yes: **Guido Nolte

Reviewer #2: No

Reviewer #3: No

---

## [Editor Report · Acceptance letter]

20 Oct 2020

PCOMPBIOL-D-20-00589R1 

Tensorpac : an open-source Python toolbox for tensor-based Phase-Amplitude Coupling measurement in electrophysiological brain signals

Dear Dr Combrisson,

I am pleased to inform you that your manuscript has been formally accepted for publication in PLOS Computational Biology. Your manuscript is now with our production department and you will be notified of the publication date in due course.

With kind regards,

Laura Mallard
